# The potential impact of declining development assistance for health on population health in Malawi: A modelling study

Margherita Molaro[1]*, Paul Revill[2], Martin Chalkley[2], Sakshi Mohan[2], Tara D. Mangal[1,2], Tim Colbourn[3], Joseph H. Collins[3], Matthew M. Graham[4], William Graham[4], Eva Janoušková[3], Gerald Manthalu[5], Emmanuel Mnjowe[6], Watipaso Mulwafu[6], Rachel E. Murray-Watson[1], Pakwanja D. Twea[5], Andrew N. Phillips[3], Bingling She[1], Asif U. Tamuri[4], Dominic Nkhoma[6], Joseph Mfutso-Bengo[6], Timothy B. Hallett[1]

**1** MRC Centre for Global Infectious Disease Analysis, Jameel Institute, School of Public Health, Imperial College London, London, United Kingdom, **2** Centre for Health Economics, University of York, York, United Kingdom, **3** Institute for Global Health, University College London, London, United Kingdom, **4** Centre for Advanced Research Computing, University College London, London, United Kingdom, **5** Department of Planning and Policy Development, Ministry of Health and Population, Lilongwe, Malawi, **6** Kamuzu University of Health Sciences, Blantyre, Malawi

* margherita.molaro@ic.ac.uk

## Abstract

### Background

Development assistance for health (DAH) to Malawi will likely decrease as a fraction of Gross Domestic Product (GDP) in the next few decades. Given the country's significant reliance on DAH for the delivery of its healthcare services, estimating the impact that this could have on health projections for the country is particularly urgent.

### Methods and findings

We use the Malawi-specific, individual-based "all diseases—whole health-system" *Thanzi La Onse* model to estimate the impact that declining DAH could have on health system capacities, proxied by the availability of human resources for health, and consequently on population health outcomes, in the period 2019–2040. We estimate that the range of DAH forecasts considered could result in a 7.0% (95% confidence interval (CI) [5.3, 8.3]) to 15.8% (95% CI [14.5,16.7]) increase in disability-adjusted life years compared to a scenario where health spending as a percentage of GDP remains unchanged. This could cause a reversal of gains achieved to date in many areas of health. The burden due to non-communicable diseases, on the other hand, is found to increase irrespective of yearly growth in health expenditure, assuming current reach, and scope of interventions. Finally, we find that greater health expenditure will improve population health outcomes, but at a diminishing rate. The main limitations of this study include the fact that it only considered gradual

which permits unrestricted use, distribution, and reproduction in any medium, provided the original author and source are credited.

**Data availability statement:** The Thanzi La Onse model is open source and available for review and usage at https://github.com/UCL/TLOmodel. In particular, the outputs analysed in this study can be reproduced from model tag "Molaro_et_al_2025_Impact_of_DAH_decline_final" (accessible at https://github.com/UCL/TLOmodel/tags), using the scenario file src/scripts/healthsystem/impact_of_const_capabilities_expansion/scenario_impact_of_capabilities_expansion_scaling.py. The scripts used to generate the plots in the manuscript can be found the same directory, in the file analysis_impact_of_capabilities_expansion_scaling.py. In addition, post-processed output data can be directly obtained from Zenodo at 10.5281/zenodo.15442201.

**Funding:** This project is funded by The Wellcome Trust (223120/Z/21/Z to TBH) and contributed to the salaries of MM, BS, and TM. MM, BS, TM, and TBH acknowledge funding from the MRC Centre for Global Infectious Disease Analysis (reference MR/X020258/1), funded by the UK Medical Research Council (MRC). This UK-funded award is carried out in the frame of the Global Health EDCTP3 Joint Undertaking. The funders had no role in study design, data collection and analysis, decision to publish, or preparation of the manuscript.

**Competing interests:** The authors have declared that no competing interests exist.

**Abbreviations:** AIDS, acquired immune deficiency syndrome; ART, antiretroviral treatment; CI, confidence interval; COPD, chronic obstructive pulmonary disease; DAH, development assistance for health; DALYs, disability adjusted life years; GBD, Global Burden of Disease; GDP, Gross Domestic Product; GoM, government of Malawi; HCWs, healthcare workers; HIV, human immunodeficiency virus; HRH, human resources for health; IHME, Institute for Health Metrics and Evaluation; NCDs, non-communicable diseases; NHE, normalised health expenditure; RMNCH, Reproductive, Maternal, Neonatal, and Child Health; RTIs, road traffic injuries; TB, tuberculosis; TLO, Thanzi La Onse; UI, uncertainty interval.

changes in health expenditure, and did not account for more severe economic shocks or sharp declines in DAH. It also relied on key assumptions about how other factors affecting health beyond healthcare worker numbers —such as consumable availability, range of services available, treatment innovation, and socio-economic and behavioural factors—might evolve.

## Conclusions

This analysis reveals the potential risk to population health in Malawi should current forecasts of declining health expenditure as a share of GDP materialise, and underscores the need for both domestic and international authorities to act in response to this anticipated trend.

## Author summary

### Why was this study done?

- Development assistance for health (DAH) is an important source of funding for public healthcare services in Malawi.

- Its contribution is expected to decline in the near future, with potential repercussions on the ability of the public healthcare system to dispense services.

- The extent of the impact this could have on population health outcomes is currently poorly understood.

### What did the researchers do and find?

- We considered different possible scenarios of health expenditure between 2019 and 2040, and estimated how they would affect the number of available healthcare workers (HCWs) over this period.

- We used the Thanzi La Onse mathematical model to simulate how each scenario of HCW availability would impact the health burden from major causes of ill health, accounting for disease dynamics and health services utilisation.

- We found that the range of possible DAH declines considered would result in an increase in the population health burden—quantified by disability adjusted life years—of 7%–15.8%, compared to the case where current health expenditure trends are sustained.

- We also found that it could lead to a reversal of important gains achieved to date in key areas of health.

### What do these findings mean?

- This study helps both domestic and international funders understand how different funding choices can affect future health outcomes in the country.

- The main limitations of this study include the fact that it only considered gradual changes in health expenditure, and did not account for the possibility of economic shocks or sharp declines in DAH.

- It also had to make assumptions about how factors that affect people's health beyond the number of HCWs available—like the availability of medical supplies, types of services offered, new treatments, and changes in living conditions or behaviour—might change over time.

## Introduction

Malawi has made remarkable progress in health in the last few decades, achieving an 18-year increase in life expectancy at birth in the country between 2000 and 2021 [1]. Many of these gains were achieved with the significant support of international donors, who contributed up to ~55% of total health expenditures in the country in 2018/2019 [2]. However, the Institute for Health Metrics and Evaluation (IHME) is forecasting a reduction in the contribution of development assistance for health (DAH) as a fraction of the country's Gross Domestic Product (GDP) in the next few decades, with the rising contribution of government expenditure on health not expected to increase fast enough to compensate for the shortfall [3].

This would happen at a crucial time for Malawi, which has only recently made substantial, yet potentially reversible, gains in major infectious disease pandemics such as human immunodeficiency virus (HIV)/acquired immune deficiency syndrome (AIDS), tuberculosis (TB), and malaria. In addition, a rise in the contribution of non-communicable diseases (NCDs) to the country's health burden means the healthcare system is facing a double burden of infectious diseases and NCDs [4,5].

In this analysis, we translate a number of hypothetical long-term health expenditure scenarios for Malawi—which may or may not realise depending on actions taken by both international and national funders and stakeholders—into equivalent levels of human resources for health (HRH) capacity. We then make use of the Malawi-specific "all diseases—whole health—system" Thanzi La Onse (www.tlomodel.org, [6]) model to estimate the health burden, quantified in disability adjusted life years (DALYs), between 2019 and 2040 under different levels of annual growth rates in health expenditure over the same period, including those forecasted by the IHME. These health burden estimates are further disaggregated by causes of illness, to understand the divergent trajectory of these causes under different levels of annual growth rates in health expenditure.

## Method

### Overview

The individual-based model *Thanzi La Onse* (TLO, www.tlomodel.org, [6]) simulates the evolution of the health burden in Malawi while capturing: its demographic growth; the evolving incidence of all major risk factors, infectious diseases, NCDs, and comorbidities; the health-seeking and treatment-adherence behaviour of those at risk of or affected by a medical condition requiring care; the range of potential interventions available to both prevent and treat those medical conditions, as well as the effectiveness of these interventions; the accuracy of referral and diagnostics; and the extent to which health services are able to meet the demand for care in all its requirements, including timely access to relevant HRH and consumables. All these factors are modelled explicitly and self-consistently, and are extensively calibrated to available data in the period 2015–2019 (see [6] for details).

The medical conditions explicitly modelled in this analysis, in particular, include major infectious diseases, such as HIV/AIDS [7,8], TB [8], and malaria [8]; NCDs, such as cancers (including bladder, breast, oesophageal, prostate, and the combined effect of all others), cardio-metabolic disorders (including Diabetes Type 2, Hypertension, Stroke, Ischaemic Heart Disease, Myocardial Infarction), chronic obstructive pulmonary disease (COPD), depression, diarrhoea, epilepsy; road traffic injuries (RTIs) [9]; and the most prevalent conditions related to maternal and newborn health [10] as well as major causes of ill health among children under five years of age, including acute lower respiratory infections [11],

measles, childhood diarrhoea, and stunting. A comprehensive, versioned documentation of all modelling assumptions for each of these conditions, including assumed risk factors and a review of all data sources and references adopted in each case, is available in the publicly-accessible repository of the model version used in this analysis, under docs/write-ups. This includes documentation (under the "Lifestyle" module) detailing how the model incorporates various social determinants of health, such as education level, wealth, marital status, and urban or rural residency. Finally, the incidence of all causes of ill health not directly modelled through an individual-based approach, accounting for approximately 19% of deaths and 28% of DALYs documented in Malawi between 2015 and 2019, is captured using Global Burden of Disease (GBD) projections (see [6] for details). More details on the model can be found in [6].

We use this model to simulate the health burden that would be incurred under different scenarios of health expenditure in Malawi between 2019 and 2040 (inclusive). Each expenditure scenario is assumed to result in an expansion of HRH in the country from capabilities initially calibrated to 2018, as discussed in detail in the next section. Because the ability of the healthcare system to meet the demand for care in the model is constrained by the HRH available, this allows us to estimate the return in health from each expenditure scenario.

Indeed, given realistic representations of **(i)** the patient-facing time available from each medical cadre at each simulated facility, and **(ii)** the time required from different medical cadres by each type of appointment, both adjusted to present-day productivity levels and available resources, the TLO model ensures that, on each day, care can only be dispensed until patient-facing time has been exhausted (see S1 Text). Failure to receive care results in an increased probability of adverse health outcomes for the individual and, in the case of infectious diseases, of further infection spread among the population, accurately capturing repercussions of HRH constraints on the overall health burden.

In this analysis, only the relationship between expenditure in HRH and health outcome is directly captured (see the Discussion section), while consumable availability is assumed to be perfect. We perform a sensitivity analysis on the consumable availability assumption in S2 Text. Finally, we note that the assumed efficacy of specific interventions is informed by data and inferred through the calibration process. The effectiveness of interventions in reality can, of course, evolve over time, influenced by factors such as the emergence of drug resistance to some infectious pathogens, advancements in treatment, or changes in policy. In this analysis, however, we do not capture the uncertainty around the future effectiveness of the treatments considered, and instead assume this to be constant with time.

**Health expenditure scenarios**

We assume that the expansion of HRH capabilities in the country matches the combined growth of GDP and fraction of GDP allocated to healthcare expenditures ($f_{HE}$) through combined government and DAH efforts. This assumption is reviewed in detail in the Discussion section.

For simplicity and to facilitate interpretability, we assume that annual fractional changes in GDP and $f_{HE}$, referred to as $g_{GDP}$ and $g_{fHE}$, respectively, are constant for the entire simulated period, although fluctuations are expected in practice. HRH capabilities available in any year $i$ can therefore be expressed relative to those in the previous year as:

$$\frac{\text{HRH}^i}{\text{HRH}^{i-1}} = (1 + g_{GDP}) \times (1 + g_{fHE}) \tag{1}$$

where therefore $[(1 + g_{GDP}) \times (1 + g_{fHE}) - 1]$ constitutes the yearly growth of the overall health expenditure, and consequently an equal growth in expenditure on HRH and patient-facing time available, and assume expansion of capabilities starts in the year $i = 2019$ (see S1 Text for details). The expansion of resources due to different health expenditure scenarios can therefore be thought of as an increase in the total amount of medical officers' patient-facing time available at each facility, resulting in the healthcare system being able to dispense a higher number of services. Initial HRH capabilities assumed in 2018 are extensively calibrated to available data, disaggregated by medical cadre, facility level, and district

[12]. In each scenario, we then assume that capabilities in each category are expanded by the same yearly percentage growth, such that the initial relative distribution of healthcare workers (HCWs) is maintained throughout the expansion period. All these assumptions are reviewed in the Discussion section.

In this analysis, the HRH expansion scenarios considered are determined by different expectations around how the GDP and $f_{HE}$ will grow with time. The range of scenarios considered are summarised and motivated in Table 1. Our worst-case scenario is one where capabilities are not expanded at all from those in 2018 ("No growth" scenario). In all other cases, we assume a fixed GDP growth per year of $g_{GDP} = 4.2\%$, which corresponds to the average annual percentage growth of GDP in Malawi (expressed in constant 2015 US$) between 1960 and 2020 according to World Bank data. Two of the scenarios considered ("<<GDP growth" and "<GDP growth") specifically approximate the lower and upper bounds in the 95% uncertainty interval (UI) reported in the IHME forecasts, as discussed in S3 Text.

Each scenario $s$ is characterised by a different normalised health expenditure (NHE) incurred over the relevant period. We define this in dimensionless units relative to HRH expenses in 2018 as:

$$NHE_s = \frac{1}{HRH^{2018}} \sum_{i=2019}^{2040} HRH_s^i$$

(2)

Finally, the health outcome of each health expenditure scenario is quantified by DALYs assuming a life-expectancy [13] of 70 years, and adopting disability adjustment factors from [14]. The initial representative population size assumed in 2010 is of 100,000 individuals, while results reported are scaled to the true population size in Malawi in 2010 of 14.5 million individuals. Each scenario was simulated 10 independent times, each with independent random draws to capture stochastic uncertainty; this specification was found to be sufficient to give stable estimates of the mean and variance of the health burden obtained under each cause included in the simulation over independent realisations. For each of the scenarios, we plot the mean and the 95% interval across the 10 model realisation. The 95% interval is a non-parametric confidence interval (CI), obtained by calculating the 2.5% and 97.5% quantiles in the simulated outcomes assuming a linear interpolation in the distribution.

## Ethics statement

The *Thanzi La Onse* project received ethical approval from the *College of Medicine Malawi Research Ethics Committee* (COMREC, P.10/19/2820) in Malawi. Only publicly available anonymised secondary data is used in the *Thanzi La Onse* model; therefore, individual informed consent was not required.

Table 1. Capabilities expansion scenarios considered. $g_{GDP}$ and $g_{fHE}$ refer to the growth rate of annual GDP (in constant 2015 US$) and growth rate of fraction of GDP allocated to health ($f_{HE}$) assumed, leading to a yearly expenditure growth per scenario of $[(1 + g_{GDP}) \times (1 + g_{fHE}) - 1]$. The normalised health expenditure (NHE) associated with each scenario (defined in Eq. 2) is also included. Scenarios capturing the lower and upper bounds in the 95% UI in the forecasts by the Institute for Health Metrics and Evaluation (IHME) are also included, and discussed in more detail in S3 Text.

| Label | $g_{GDP}$ (%) | $g_{fHE}$ (%) | Yearly expenditure growth (%) | Normalised Health Expenditure (NHE) | Description |
|---|---|---|---|---|---|
| No growth | 0 | 0 | 0 | 22.00 | No expansion of capabilities |
| <<GDP growth | 4.2 | −3.0 | 1.1 | 24.93 | IHME forecast (lower-bound) |
| <GDP growth | 4.2 | −1.5 | 2.6 | 30.08 | IHME forecast (upper-bound) |
| GDP growth | 4.2 | 0 | 4.2 | 36.53 | Expansion follows GDP growth, with $f_{HE}$ fixed at the implicit 2018 value. Captures the current level of healthcare expenditure. |
| >GDP growth | 4.2 | +1.5 | 5.8 | 44.60 | Growth above GDP |
| >>GDP growth | 4.2 | +3.0 | 7.3 | 54.75 | Growth significantly above GDP |

## Results

### How does health scale with expenditure on human resources for health?

In Fig 1a we show the evolution of yearly DALYs incurred under different hypothetical scenarios of health expenditure, including all causes listed in the Methods section and the correction from GBD projections from all remaining causes not explicitly included, as discussed in the same section. A steady rise in yearly DALYs incurred is observed in the long-term for all scenarios considered, suggesting that an expenditure above the largest yearly expenditure growth considered in this analysis (">>GDP growth") would be required to stabilise the long-term health burden in the country. This is partly driven by population growth, as illustrated by the equivalent evolution of the life expectancy in Fig 1b: the lowest level of expenditure considered ("No growth" scenario) is indeed still able to stabilise the average life expectancy over the whole period, however, failing to achieve any improvements in individual outcomes.

In Fig 2a, on the other hand, we show the total DALYs incurred between 2019 and 2040 as a function of the yearly health expenditure growth, as well as the normalised total health expenditure (NHE, defined in Eq. 2) characterising each expenditure scenario. In Fig 2b, on the other hand, we show the percentage of total DALYs averted compared to the "GDP growth" scenario, which captures the current level of health expenditure.

While the reduction in health burden initially achieved is around ~10 million DALYs for every percentage point increase in expenditure growth, this trend becomes sublinear as the overall expenditure increases above ~4%, eventually flattening around 8%–9%. In particular, a reduction in $f_{HE}$ with rates in the 95% UI of the IHME forecast (shown by the blue shaded area in the figure) would result in an excess of between 7.0% (95% CI [5.3, 8.3]) and 15.8% (95% CI [14.5,16.7]) increase in total DALYs incurred compared to a fixed $f_{HE}$ (i.e., current level of health expenditure) scenario ("GDP growth" scenario)

### How are key areas of health affected?

In Fig 3, we show how three important areas of health are affected by different expenditure strategies, namely:

i)   major infectious diseases primarily supported via vertical programmes, namely HIV/AIDS, TB, and malaria (HTM);

ii)  Reproductive, Maternal, Neonatal, and Child Health (RMNCH), including lower respiratory infections, childhood diarrhoea, maternal disorders, measles, and neonatal disorders; and

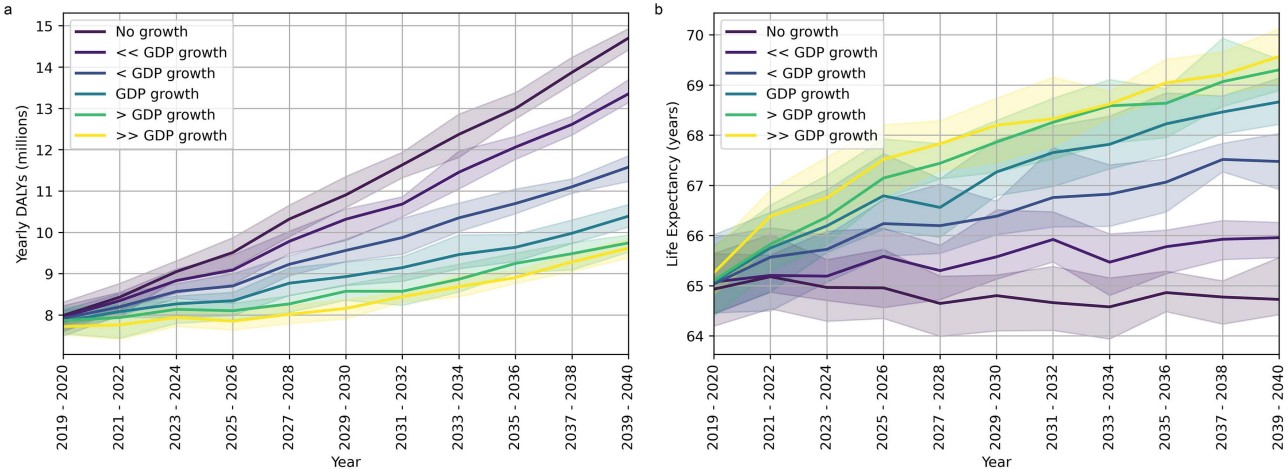

**Fig 1. Evolution of health burden and life expectancy under different expenditure scenarios. (a)** Total yearly DALYs (averaged over two-year periods) incurred under different expenditure scenarios. **(b):** Life expectancy (averaged over two-year periods) achieved under different expenditure scenarios. In both plots, solid lines represent mean values, while shaded areas indicate the 95% CIs defined in the Methods section.

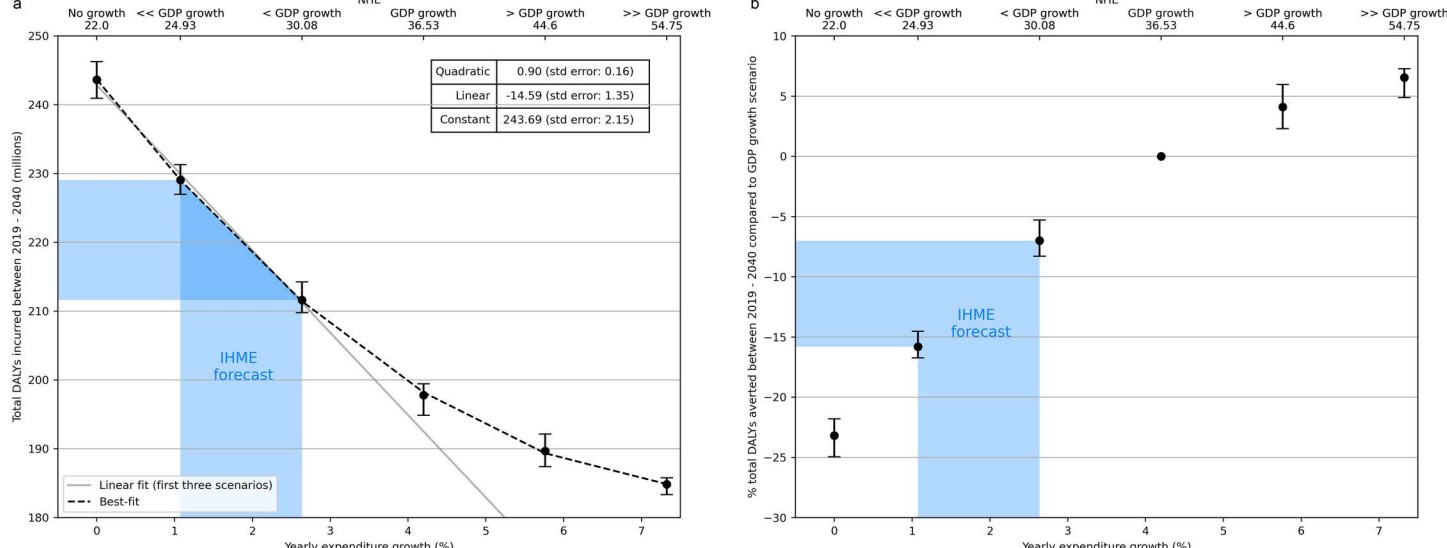

**Fig 2. Scaling of overall health burden with yearly expenditure growth. (a)** Total DALYs incurred in the period 2019–2040 (inclusive) as a function of the yearly expenditure growth, as well as the normalised total expenditure (NHE) over that period under each scenario (top x-axis), as defined in Eq. 2. The best-fit parameters for the function shown are summarised in the table inside the plot, while the linear best-fits to the first three levels of yearly expenditure growth considered are only included for visual guidance. **(b)** Percentage DALYs averted in the same period compared to the "GDP growth" scenario, where this captures the current level of health expenditure as a fraction of GDP. The blue shaded area shows the 95% UI in the IHME forecast. In both plots, points represent mean values, while error bars indicate the 95% CIs defined in the Methods section.

iii) NCDs, including COPD, cancers, depression/self-harm, diabetes, epilepsy, heart disease, kidney disease, and stroke, alongside RTIs.

Each panel shows the average yearly DALYs lost under each area of health between 2019 and 2040 as a function of the yearly expenditure growth associated with that scenario, while the horizontal lines show the yearly DALYs incurred due to that cause in 2018, the year before scenarios start diverging. This means that if the DALYs incurred under a scenario fall below the respective horizontal line, an average decline in the health burden from that cause was achieved over the period for that scenario. On the other hand, if the DALYs incurred lie above the horizontal lines, the expenditure scenario led to an average increase in the health burden due to that cause between 2019 and 2040. A breakdown of the time evolution of the individual burden from these health conditions is additionally included in S4 Text.

A downward trend in DALYs incurred due to HIV/AIDS appears to be achievable by all expenditure scenarios, despite significant population growth over this period, suggesting that the important gains made in this area in previous years can be sustained by existing HRH capabilities—although the increase in average yearly DALYs with diminishing yearly expenditure growth suggests that such gains may still be reversed if existing HRH capabilities were to be reduced. However, this finding is dependent on the assumed availability of relevant medical consumables and the relatively short time-scale being examined; the latter may indeed be too brief for a resurgence in HIV incidence to significantly increase the demand for ART and the overall HIV/AIDS health burden, as discussed in detail in the Discussion section. The rise in malaria and TB burden would also appear to be mostly contained, but requiring a minimum of "GDP-growth" level expenditure (therefore above IHME projections) to be at least stabilised.

On the other hand, DALYs due to RMNCH would, under IHME projections, significantly rise compared to their 2,018 level over the period considered, mainly driven by the population growth over this period. An increase in HRH availability

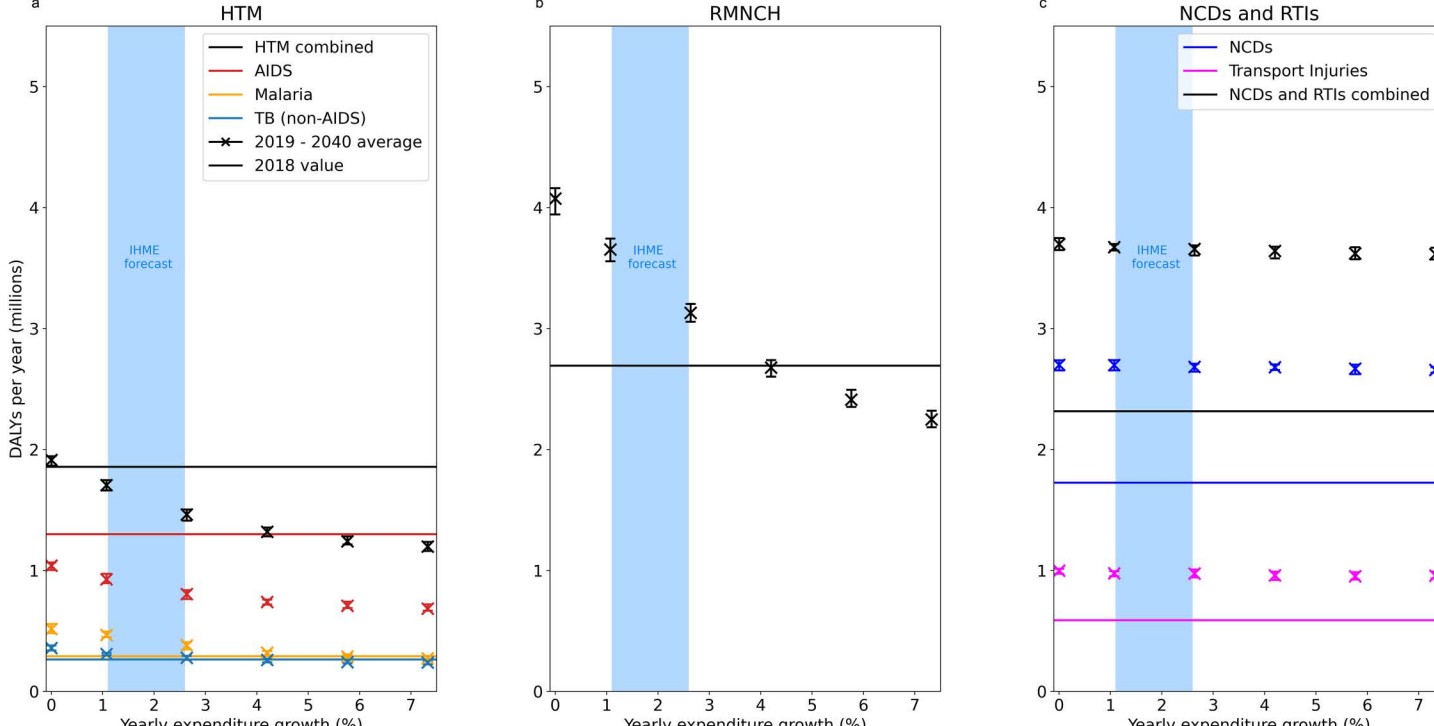

**Fig 3. Impact on different areas of health.** Average yearly DALYs incurred between 2019 and 2040, grouped into three meaningful categories: HTM, including HIV/AIDS, TB, and malaria **(a)**; RMNCH, including lower respiratory infections, childhood diarrhoea, maternal and neonatal disorders, and measles **(b)**; and NCDs, including COPD, cancers, depression/self-harm, diabetes, epilepsy, heart and kidney disease, and stroke, alongside RTIs **(c)**. In the case of HTM, DALYs in this area are additionally shown broken down by individual causes. In all plots, points represent mean values, while error bars indicate the 95% CI defined in the Methods section. Horizontal lines indicate the yearly DALYs burden for each cause in 2018. This means that any scenario *above* the respective 2,018 level incurred, on average, a worsening of the health burden due to that cause over the 2019–2040 period, whereas any scenario *below* the respective 2,018 level incurred an improvement. Finally, the blue shaded area shows the 95% UI in the IHME forecast. A break-down of the time evolution of the burden due to all of these causes of ill health individually is included in S4 Text.

in line with 'GDP growth" and above appears to be effective at containing this rise and leading, for expenditures above "GDP growth", to a downward trend in DALYs incurred in this area of health despite an increase in population size.

Finally, and unlike for other areas of health, the rising health burden due to NCDs appears to be largely unaffected by the availability—and therefore expansion—of HRH capabilities. This is due to the complex challenges currently preventing effective NCD programme delivery in the country [15–17]. Notice that this unconstrained rise in NCDs is therefore what drives the long-term rise in the overall health burden even in the ">>GDP growth" expenditure scenario, which can be observed in Fig 1.

## Discussion

We have estimated the trends in long-term health-burden that can be expected in Malawi under different scenarios of healthcare expenditure in the period 2019–2040, approximated by an equivalent HRH capabilities expansion. We found that the decline in total DALYs incurred between 2019 and 2040 is initially close to ~10 million DALYs per percentage point increase in yearly expenditure growth, but exhibits diminishing returns above 4%. Reasons for these diminishing returns are independent of assumptions around consumable availability (see S2 Text) and likely linked to: the assumed fixed range of preventive, screening, and treatment services offered by the healthcare system over the entire period, which could instead be expanded to include less cost effective options once the most urgent medical needs in the

population have been met; persisting difficulties in health service access by certain sections of the population; imperfect diagnostic and referral accuracy; and the intrinsic (less than perfect) effectiveness of each intervention, which are all factors explicitly captured in the TLO model.

This showed that if the 95% UI in the IHME's projected decline in the fraction of GDP allocated to healthcare from combined governmental and international aid efforts should realise, the country would incur a percentage increase in total DALYs lost of between 7.0% (95% CI [5.3, 8.3]) and 15.8% (95% CI [14.5, 16.7]) compared to current levels of health expenditure, and that the significant gains made by Malawi in the past in important areas of health such RMNCH, malaria, and TB could be reversed. Sustaining or exceeding current levels of health expenditure, on the other hand, would ensure the burden due to these causes can continue to decrease over time despite population growth. In the case of HIV/AIDS, progress made to date—which includes the country already meeting two of the three targets set out by USAIDS for 2030 [18]—appears to have made the decline in this burden sustainable by current HRH capabilities.

Finally, the significant rise in the health burden due to NCDs was found to be unaffected by the level of expenditure in HRH, suggesting that expanding the reach and scope of preventive, screening, and treatment services offered would play a key role in ensuring that health investments can effectively translate into a reduction of the health burden due to these causes of ill health in the future [15–17].

This modelling approach enables a health system-wide estimation of the potential impact of different hypothetical health system funding scenarios on population health, using a model extensively calibrated to the local context. The Malawian Ministry of Health has been closely engaged from the earliest stages of the TLO model's development and the conception of this analysis. As a result, this work provides valuable, context-sensitive scientific evidence [19] to support the wider and complex deliberative process of public healthcare policy design [20,21]

The strength of this modelling approach indeed lies in its calibration to Malawi-specific factors—including population properties, healthcare structure, and epidemiological landscape—which ensures a country-tailored and contextually relevant analysis. The generalisability of these findings beyond Malawi, therefore, is unknown. However, by highlighting the potential consequences on health outcomes of reductions in DAH for Malawi, we hope to underscore the urgency of conducting similar analyses in other countries in the region and beyond that also rely heavily on foreign aid, and therefore are similarly vulnerable to its decline.

To our knowledge, no other study has directly linked future healthcare spending implications to population health outcomes in Malawi or other countries; we are therefore unable to conduct a comparative exercise on our results at this time.

This analysis relied on a number of key assumptions. With regards to health expenditure projections, the significant uncertainty surrounding future trends in GDP, DAH, and $f_{HE}$, especially over extended periods of time, is inevitably reflected by any long-term projections assumed in this analysis. To address this, our analysis therefore deliberately considered a wide range of possible yearly expenditure growth scenarios, and remained agnostic as to which is more likely to be realised in the future, focussing instead on quantifying the population-health consequences in each case. For example, while the range of $g_{fHE}$ values considered included IHME projections, this was deliberately extended to capture a wider range of potential annual changes in $f_{HE}$.

We further emphasise that in this analysis, it is the assumed annual expenditure growth—which reflects the combined effects of $g_{GDP}$ and $g_{fHE}$—which determines the HCW capacity expansion. This allows for flexibility in interpreting each scenario under different $g_{GDP}$ and $g_{fHE}$ assumptions, by adjusting one parameter while proportionally rescaling the other. For example, the "<GDP growth" scenario ($g_{GDP}$=4.2%, $g_{fHE}$=−1.5%) could equally represent a scenario where $g_{GDP}$=2% and $g_{fHE}$=0.62%. And while in designing these scenarios, we assumed that these parameters could vary independently of one another, each scenario could therefore easily be reinterpreted as reflecting any level of correlation between them, which may better reflect how these two parameters evolve in reality.

Finally, we assumed that $g_{GDP}$ and $g_{fHE}$ remain constant throughout the simulated period for each scenario. In doing so, we have chosen to prioritise interpretability over complexity: accounting for fluctuations in $g_{GDP}$ and $g_{fHE}$ over the simulated

period would have introduced complex temporal dynamics in the disease burden, influenced by the timing, magnitude, and cumulative effect of prior investment, as well as the intrinsic transmission dynamics of the simulated infectious diseases. Such interactions would have significantly complicated result interpretation and scenario comparison, particularly given the arbitrary nature of the timing and magnitude of any simulated fluctuations. We therefore believe this to be a reasonable assumption at present, and postpone a more detailed analysis of the effect of these fluctuations to a later study.

In assuming that the capabilities expansion considered in this analysis scaled simply with GDP and $f_{HE}$ growth, we implicitly assumed that the fraction of the total health expenditure (GDP × $f_{HE}$) allocated to HRH (as opposed to new infrastructure, the purchasing of consumables and equipment, administrative costs, etc.) is constant with time, while the *real costs* of HRH are also assumed to be constant, and that the relative distribution of HCWs across different cadres, facility levels, and districts remains constant as HRH capabilities are expanded. This strategy for resource allocation may be far from optimal. A more targeted expenditure in particular districts/facility levels/types of cadres may indeed result in a higher return in health from the same expenditure. This analysis should therefore be seen as a "lower bound" to the health benefit that could be obtained from the same expenditure.

In addition, we assumed that any expenditure – based on the above-listed assumptions—directly translated into an expansion of available patient-facing time. In this simplified approach, we therefore did not account for (i) possible variations in HCW productivity due to the expansion of existing cadres or other influencing factors, and (ii) the fact that the realisation of such scenarios would require adequate and timely expenditures in training and recruitment of additional HCWs, as well as potentially the expansion of existing infrastructures, facilities, and equipment to accommodate the intake of new personnel. These costs may amount to a higher share of the healthcare budget currently allocated to these areas; fully capturing them while imposing a cap on overall expenditure may therefore limit the amount of HRH capabilities expansion achievable under the same assumed GDP and $f_{HE}$ growth, as would potential wage increases over the simulated 22-year period. Once such additional costs are captured, such considerations could, however, easily be embedded in this analysis by refactoring the assumed growth/total expense incurred under each scenario considered. Accounting for variations in healthcare worker productivity would also be an extremely important addition to this analysis; while this is a focus of future work on the model, more empirical analysis is first required to better understand how productivity of healthcare workers varies as a function of environmental factors (such as overall demand for care and number of healthcare workers available), policy decisions, incentives, quality of management, and improvements in available technologies.

Furthermore, while HRH constitutes one of the most important constraints to healthcare delivery and universal healthcare access [22,23] we did not capture how other constraints—such as consumables, ambulances, and others—would scale with increasing expenditure, nor how factors beyond health system provision which may impact the country's health outcomes, such as access to improved (or declining) infrastructure, education, and employment, would evolve with time, or themselves be influenced by public healthcare spending. While in this analysis we assumed perfect consumable availability, its main conclusions were, however, found to be unchanged under an assumption of present-day consumable availability (see S2 Text). In addition, all health services were assumed to be competing equally for the same limited resources. In reality, different funding streams may result in a higher share of overall capabilities being reserved for specific programmes, e.g., through vertical funding ([24]). This share, however, would likely evolve over the period considered as a result of a decline in the contribution of DAH.

In this analysis, we were agnostic as to how much of the assumed investment into HRH would be contributed by DAH versus the government of Malawi (GoM). The GoM currently covers the highest proportion of HRH spending (68%) [25]. Health worker salaries and benefits account for 70% of this cost, and are covered at 90% by the GoM. Assuming this present-day allocation of costs, it may be argued that a contraction in international donors support may not affect HRH as significantly as assumed in these scenarios. It is, however, not unreasonable to assume that the GoM may be forced to reallocate some of its HRH funding were international donors to withdraw their support in other areas of health, resulting in an equivalent contraction. Furthermore, the financial strain placed by a decline in donor support could have a detrimental

impact on staffing levels, medical consumable supply, and technical capacity, as experienced in other countries which went through a similar funding transition [26]. Our analysis does not capture these potential effects.

Finally, this analysis did not capture the contribution of prepaid private and out-of-pocket health expenditures to the overall health outcome of the population. The IHME forecasts that the relative contribution of these types of health expenditures to the overall health expenditure in the country will grow over this period from an estimated 17% in 2019 [3]; however, the future trajectory of prepaid private and out-of-pocket health spending is itself likely to be influenced by the level of investment in public healthcare, as changes in public provision may shape demand for private alternatives.

This analysis further assumed perfect consumable availability in all disease areas throughout the entire simulated period. While a more nuanced analysis of consumable availability under varying funding scenarios would be highly valuable, we refrained from including it at this stage due to the complexity of accurately modelling the relationship between financial investment and improvements in consumable availability: indeed, the costs associated with strengthening supply chains, ensuring effective stock management, and minimising shortages, remain at present poorly understood.

A sensitivity analysis on this assumption was conducted by testing a scenario in which consumable availability instead remains fixed at its present-day level [6] without any improvement (see S2 Text). It is, however, important to emphasise that for mostly vertically funded programmes such as HIV/AIDS, current levels of relevant consumable availability in Malawi are already extremely high [7]: in 2018, the year used for present-day consumable calibration, the average availability of adult antiretroviral treatment (ART) and HIV tests across different facility levels was as high as 94% and 88.5% respectively (see [7], Fig 2 therein), an availability generally much greater than that for consumables related to other disease areas [6].

As discussed in the Results section, in this analysis, the health burden due to HIV/AIDS was found to decline under all investment scenarios considered despite population growth over the period considered (2019–2040), both under an assumption of perfect consumable availability, and one of present-level consumable availability (see S2 Text). While promising, we therefore seek to emphasise that this finding of resilience of the HIV/AIDS burden to a lack of investment into the healthcare workforce is highly contingent on a high availability of relevant consumables currently achieved in the country, and as such could be severely compromised by any decline in this availability.

Furthermore, we note that the time frame considered in this analysis (2019–2040) may be too short to capture any negative effects on the metric of choice, DALYs. Any increase in the underlying incidence of the disease as a result of a lack of expansion of HWC may indeed only become detectable as an increase in the health burden on longer timescales. This is due to the fact that a high proportion of HIV-infected individuals in Malawi are already receiving ART and are virally suppressed: in 2022, as many as 93% of people living with HIV in Malawi were aware of their HIV status; among those diagnosed, 97% were receiving ART, and of those on ART, 93% had achieved viral suppression [27]. Once off treatment, individuals who are virally suppressed may remain asymptomatic for years, depending on their health when ART was initiated. However, they would immediately become infectious again; this, together with reduced access to prevention and ART services, would lead to a resurgence in HIV infections, which may, however, only manifest as an increase in DALYS beyond the time period of this study.

Finally, we seek to clearly state that none of the scenarios considered in this analysis reflect a rapid reduction in HIV/AIDS funding, which would lead to an immediate rise in AIDS deaths and HIV incidence [28,29]. First, such defunding would represent a contraction of current resources, i.e., a negative yearly expenditure growth, which is beyond the range considered in our analysis. Second, it would create a chaotic disruption to health system provision —as services are redesigned and staff redeployed—which stands in contrast to the gradual, and optimally-managed changes modelled here.

Lastly, we acknowledge that a number of important assumptions were made with regard to the future incidence of NCDs, which will be shaped by a large number of complex factors [4]. The model explicitly accounts for many of these, such as level of wealth, education, sugar, salt, alcohol intake, and body mass index, and does so by extrapolating current trends on the evolution of the incidence of these factors into the future. (Details on how these factors are accounted

for in the modelling of each individual disease can be found in the model's documentation, as discussed in the Methods section). Of course, significant uncertainty remains around such extrapolations, as well as uptake and effectiveness of treatment. By highlighting how significant the future health-burden from such causes would be should the current trends be maintained, this analysis highlights the urgency of implementing preventive as well as curative strategies to ensure both the incidence and the health-cost from these NCDs can be contained in the future.

In conclusion, this analysis is the first, to our knowledge, to quantify the potential overall consequences of a number of potential health expenditure scenarios in Malawi in the future, described by an equivalent expansion of HRH. It demonstrated the potential risk of reversing gains in several key areas of health in the country if current forecasts on the decline of contribution from development assistance for health were to be realised, and highlighted the need for domestic and international authorities to act in response to this predicted trend. In particular, it found that current levels of expenditure on HRH as a fraction of GDP should be sustained in order to prevent a reversal of gains in many important areas of health, such as RMNCH, TB, and malaria. Furthermore, it found that the rising contribution of NCDs to the national health burden is currently largely unaffected by investments in HRH, suggesting that an expansion to the scope and reach of preventive and treatment programmes relevant to these diseases should be considered, if the rising contribution of these conditions is to be contained.

## Supporting information

**S1 Text. Modelling HRH constraints.**
(DOCX)

**S2 Text. Effect of consumable availability.**
(DOCX)

**S3 Text. IHME forecasts of health expenditure.**
(DOCX)

**S4 Text. Time evolution of individual causes of ill health.**
(DOCX)

## Author contributions

**Conceptualisation:** Margherita Molaro, Paul Revill, Martin Chalkley, Timothy B. Hallett.

**Data curation:** Margherita Molaro.

**Formal analysis:** Margherita Molaro.

**Funding acquisition:** Timothy B. Hallett.

**Investigation:** Timothy B. Hallett.

**Methodology:** Margherita Molaro, Paul Revill, Martin Chalkley, Sakshi Mohan, Tara D. Mangal, Timothy B. Hallett.

**Resources:** Paul Revill, Martin Chalkley.

**Software:** Margherita Molaro, Sakshi Mohan, Tara D. Mangal, Joseph H. Collins, Matthew M. Graham, William Graham, Eva Janoušková, Emmanuel Mnjowe, Watipaso Mulwafu, Timothy B. Hallett.

**Supervision:** Timothy B. Hallett.

**Validation:** Margherita Molaro.

**Visualisation:** Margherita Molaro.

**Writing – original draft:** Margherita Molaro, Paul Revill, Martin Chalkley, Sakshi Mohan, Tara D. Mangal, Tim Colbourn, Joseph H. Collins, Matthew M. Graham, William Graham, Eva Janoušková, Gerald Manthalu, Emmanuel Mnjowe, Watipaso Mulwafu, Rachel E. Murray-Watson, Pakwanja D. Twea, Andrew N. Phillips, Bingling She, Asif U. Tamuri, Dominic Nkhoma, Joseph Mfutso-Bengo, Timothy B. Hallett.

**Writing – review & editing:** Margherita Molaro, Paul Revill, Martin Chalkley, Sakshi Mohan, Tara D. Mangal, Tim Colbourn, Joseph H. Collins, Matthew M. Graham, William Graham, Eva Janoušková, Gerald Manthalu, Emmanuel Mnjowe, Watipaso Mulwafu, Rachel E. Murray-Watson, Pakwanja D. Twea, Andrew N. Phillips, Bingling She, Asif U. Tamuri, Dominic Nkhoma, Joseph Mfutso-Bengo, Timothy B. Hallett.

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
