## [Editor Report · Decision Letter 0]

8 Oct 2024

Dear Dr Molaro,

Thank you for submitting your manuscript entitled "The potential impact of declining development assistance for healthcare on population health: projections for Malawi" for consideration by PLOS Medicine.

Your appeal has been considered by the PLOS Medicine editorial staff and by an academic editor with relevant expertise, and I am writing to let you know that we would like to send your submission out for external peer review.

Before we can send your manuscript to reviewers, we need you to complete your submission by providing the metadata that is required for full assessment. To this end, please login to Editorial Manager where you will find the paper in the 'Submissions Needing Revisions' folder on your homepage. Please click 'Revise Submission' from the Action Links and complete all additional questions in the submission questionnaire.

Please re-submit your manuscript within two working days, i.e. by Oct 10 2024 11:59PM.

Feel free to email me at lgaynor@plos.org if you have any queries relating to your submission.

Kind regards,

Louise Gaynor-Brook, MBBS PhD

Senior Editor

PLOS Medicine

---

## [Decision Letter · Decision Letter 1]

29 Jan 2025

Dear Dr Molaro,

Many thanks for submitting your manuscript "The potential impact of declining development assistance for healthcare on population health: projections for Malawi" (PMEDICINE-D-24-03289R1) to PLOS Medicine. Please accept my apologies for the delay in coming to a decision. The paper has been reviewed by subject experts and a statistician; their comments are included below and can also be accessed here: [LINK]

As you will see, the reviewers thought your manuscript addresses an important research question, but they also had some major concerns. Specifically, they asked for more explanation and discussion of the model, as well as more contextualisation to generalize these findings. After discussing the paper with the editorial team and an academic editor with relevant expertise, I'm pleased to invite you to revise the paper in response to the reviewers' comments. We plan to send the revised paper to some or all of the original reviewers, and we cannot provide any guarantees at this stage regarding publication.

We ask that you submit your revision by Feb 19 2025 11:59PM. However, if this deadline is not feasible, please contact me by email, and we can discuss a suitable alternative.

Don't hesitate to contact me directly with any questions (lgaynor@plos.org).

Best regards,

Suzanne

Suzanne de Bruijn, PhD

Associate Editor, PLOS Medicine

Sbruijn@plos.org

on behalf of

Louise

Louise Gaynor-Brook, MBBS PhD

Senior Editor

PLOS Medicine

lgaynor@plos.org

Comments from the academic editor:

The research question is important, but there are limitations in the manuscript in its current form.

1. In its current form, there is limited generalisability outside of Malawi. Malawi also has certain unique population, healthcare, and economic features which mean that findings could only be generalised to other countries in the region or globally with extreme caution. Considerable editing and contextualisation in the introduction and discussion would help here.

2. The authors should work to improve accessibility and clarity in their reporting, to ensure the broad audience of PLOS Medicine can understand what has been done, and why it is important.

3. Reporting of methods and results are very brief currently, and would need to be substantially expanded to allow readers (and particularly general medical/public heath readers not in the field of development/development economics) to appraise the quality and rigour of the work that has been done.

4. In particular, some model inputs were not clear to me. For example, the impact on NCDs was mentioned, but it was not clear from this manuscript which NCDs contributed data. Trauma and road traffic injuries are a major contributor to ill health and healthcare utilisation in Malawi, particularly amongst the working age population. But I didn’t see whether they were counted as contributors. To be clear, this is only one example, and I could think of many others (eg air pollution as a “risk factor”, among others). The set of risk factors and conditions included may be comprehensive as the authors state, but I couldn’t find this information in the manuscript - currently there is only a link to another pre-print reference. The methods and sources of data for this manuscript should be described in full, and stand on their own.

5. The authors should provide greater critique of sources of input data. Particularly they should discuss whether GDB estimates (given the GBD’s input data) have sufficient validity and accuracy for Malawi to justify the level of precision of their estimates.

6. The authors make a strong call in the discussion section for this modelling approach to be more widely used in development and healthcare policy. However, they do not reflect on the ethical issues around resource allocation based on these outputs. Allocation based on modelling like this may or may not be “fairer” than current approaches, but the authors should at least tackle this question in the Discussion. Specifically, from the results of their research, what would they recommend the international donor community and Malawi ministry of health fund differently over the next 10-20 years, and what would the likely consequences (positive and negative) of this be? What if this approach was applied in other countries in the region? Right now, (somewhat simplistically) conclusions reads as “we developed a very sophisticated way of predicting impact of funding, and countries should use it”.

Comments from the reviewers:

Reviewer #1: The Thanzi La Onse (TLO) model offers an innovative framework to simulate various health expenditure scenarios, yielding valuable insights into the projected health burden and resource needs within Malawi's healthcare system. However, despite the model's comprehensive approach, specific methodological aspects require enhanced transparency and rigor to ensure greater clarity and reliability in the findings. My comments are as in the following.

Major comments:

a) Several assumptions are made regarding the expansion of human resources for health (HRH) based on GDP growth rates. These assumptions need further justification, particularly the assumption of constant GDP and fraction of GDP allocated to health (fHE) over the 2019-2040 period. Given the volatility in these figures, assumptions may need to be adapted or justified with additional context.

b) The manuscript notes an assumption of perfect consumable availability throughout the model's projections. Although a sensitivity analysis is presented for this assumption, it would be beneficial to integrate more nuanced scenarios (e.g., variable consumable availability in response to funding changes) to improve the model's realism and applicability.

c) The TLO model is complex, capturing various factors affecting health in Malawi. Yet, there is limited discussion on the specific calibration of model parameters to real-world data (2015-2019). Including a summary of how well the model's outputs match historical data would strengthen its validity and improve transparency.

d) The model enforces HRH constraints by limiting healthcare delivery based on available patient-facing time, yet does not fully account for productivity variations over time.

e) The model appears to assume uniform availability of HRH across all districts and facility levels, which may not be reflective of reality. More granular modelling, with variations in HRH distribution and healthcare access across different regions, would increase accuracy.

f) While the manuscript reports 95% confidence intervals for projections (e.g., DALYs), there is limited detail on how these intervals were computed. Clarification on the statistical methods used to calculate the intervals, including any resampling or bootstrapping techniques, would strengthen the reliability of these estimates.

g) Although the model has undergone internal validation, it lacks external validation against other health projection models or empirical data. Benchmarking projections against independent data sources or similar models could add credibility and confidence in the findings.

h) The manuscript presents different growth scenarios for health expenditure, but additional statistical comparisons between scenarios would be valuable. For instance, a quantitative comparison (e.g., through effect size calculations or relative risk reduction) of health outcomes across scenarios could help quantify the impacts of varying health expenditure levels more precisely.

i) Although the model is open-source, detailed documentation on data inputs and assumptions in the supplemental materials would support reproducibility. Clear data availability statements, particularly for health outcome and HRH projections, could enhance transparency.

Additional comments:

j) The model assumes consistent effectiveness of interventions throughout the projection period. Yet, changes in intervention efficacy (e.g., due to drug resistance, policy shifts) can impact outcomes. Incorporating variable effectiveness rates or sensitivity analyses on intervention efficacy would enhance the model's adaptability to real-world conditions.

k) While the model assesses the DALYs for general categories (e.g., HIV/AIDS, NCDs), the aggregation of different health conditions within these broad categories limits the granularity of the findings. Further breakdown of health outcomes could allow for more targeted conclusions, particularly given the differing health service requirements across diseases like diabetes versus COPD within the NCDs category.

l) While consumable availability is addressed as either perfect or present-day, this binary approach may not capture the complexity of real-world supply chain challenges. A more nuanced approach, considering gradual improvements or degradation in consumable supply with changing funding levels, would provide a more granular view of system constraints.

m) While the TLO model captures demographic growth and disease incidence, it does not appear to fully account for social determinants of health or regional disparities within Malawi. Including these variables or discussing their potential impact on projections would improve the model's applicability to varied demographic settings within the country.

n) The manuscript would benefit from discussing projection accuracy metrics, such as mean absolute error (MAE) or root mean square error (RMSE), based on past health outcomes.

o) The manuscript could enhance its scenario comparison by quantifying the differences between them with confidence intervals. This would help in clearly illustrating the statistical significance (or lack thereof) between the health outcomes across different expenditure scenarios.

p) A more detailed description of the data sources used to populate the model would improve transparency. This should include the quality, limitations, and temporal range of each dataset, particularly for health burden and expenditure data.

Reviewer #2: This article reports the results from a simulation modeling exercise focused on the potential impact of decreases in health spending on the health burden in Malawi. The authors are to be commended for applying their skill set and methodology on a relevant and important case study. The results from the study provide useful insights for stakeholders in the health sector in Malawi on the implications of various scenarios of health spending that could potentially materialize. It therefore offers policymakers and decision makers in the sector possible future outcomes that can be targeted with current actions. I provide some additional comments below to further strengthen the current manuscript.

Feedback

- Recommend adding text in the introduction or discussion to clearly highlight that the results highlighted in the study are based on hypothetical scenarios which may or may not occur depending on actions that are taken. Thus, emphasizing the importance of evidence-based decision making for readers and stakeholders in the sector.

- While the simulation platform's original documentation is referenced, I recommend a summary paragraph on the model components to orient readers who may be encountering this model for the first time.

- It may be useful for the readers to add text for why the IHME dataset rather than other alternative sources of this data is utilized in the study and also why the particular dataset is utilized and not the updated versions of the IHME dataset.

- Recommend adding a paragraph to the discussion that discusses the results from this exercise in context with either any relevant studies that discuss the implication of future spending and health outcomes in Malawi or perhaps in comparable neighboring countries.

- You do this a bit in the section where you highlight the assumptions embedded in the choices available in the simulation model set up but I recommend also adding a paragraph to the discussion in which you discuss the other factors beyond human resources that impact disease burden covering factors in the health sector such as the availability of medicines etc. but also factors outside the health sector such as sanitation, employment etc. that impact health outcomes.

Reviewer #3: This manuscript presents an interesting approach with innovative modeling and data analysis. To enhance its clarity and comprehensiveness, it would be beneficial to expand on the following areas:

* Methods:

o Provide more details on the scenarios presented in Table 1, specifically explaining how key parameters (e.g., gGDP, gfHE) interact. Are these variables simultaneously determined at certain levels?

o Does the model assume a constant productivity level for the health workforce, considering the projected increase in the number of health workers.

*Results:

o Further elaboration on the modeling results related to HRH capacity is recommended, as this is a critical component of the theory of change presented. Specify the number of health workers required under each scenario and assess whether these projections are realistic given the country's existing capacity to train and deploy health workers, as well as current health labor market entry patterns. Consider discussing potential constraints and their implications for future health expenditures.

* Implications:

o Outline specific policy recommendations derived from the analysis and results.

---

* Please upload any figures associated with your paper as individual TIF or EPS files with 300dpi resolution at resubmission; please read our figure guidelines for more information on our requirements: http://journals.plos.org/plosmedicine/s/figures. While revising your submission, please upload your figure files to the PACE digital diagnostic tool, https://pacev2.apexcovantage.com/. PACE helps ensure that figures meet PLOS requirements. To use PACE, you must first register as a user. Then, login and navigate to the UPLOAD tab, where you will find detailed instructions on how to use the tool. If you encounter any issues or have any questions when using PACE, please email us at PLOSMedicine@plos.org.

* [EDITOR: CHECK FINANCIAL DISCLOSURES, COI, DAS, AND ETHICS STATEMENTS AND INCLUDE ANY NECESSARY REQUESTS]

* Please ensure that the study is reported according to the [XXXX] guideline and include the completed [XXXX] checklist as Supporting Information. When completing the checklist, please use section and paragraph numbers, rather than page numbers. Please add the following statement, or similar, to the Methods: "This study is reported as per [XXXX] guideline (S1 Checklist)."

FIGURES AND TABLES

SUPPLEMENTARY MATERIAL

REFERENCES

[STUDY TYPE-SPECIFIC REQUESTS - DELETE SECTIONS AS NECESSARY]

RCTs [REFER TO RCT CHECKLIST AND MEETING NOTES FOR DETAILS TO ADD]

* PLOS Medicine requires that all trials be prospectively registered in one of registries recognized by WHO. Please ensure that study registration details are included in the Methods section.

* Please structure the Methods section using the following sub-headings: Study design and participants, Randomization and masking, Procedures, Outcomes, Statistical analysis.

* The following outcomes measures [ADD DETAILS AS NEEDED OR DELETE BULLET POINT] appear to differ between the submitted manuscript and the protocol [and/or trial registry]. Please clarify and explain all discrepancies between the paper and protocol. If the outcomes were not prespecified in the protocol, please define them in the Methods (Outcomes section) as post hoc and explain why they were added. Post-hoc comparisons should be presented as hypothesis generating rather than conclusive.

* Please ensure that all prespecified outcomes (primary, secondary, and exploratory) are listed in the Methods/Outcomes section and indicate whether there are outcomes that are not presented in the current report.

* Please specify the dates (Month Day, Year) during which study enrollment and follow up occurred.

* Please include absolute numbers wherever you report percentages; eg, n/N (%)

* Please present the safety data for the study including numbers of specific events and whether or not adverse events are thought to be related to treatment. AEs should be reported in the abstract, per CONSORT and CONSORT-Harms.

* Please complete the CONSORT checklist (https://www.equator-network.org/reporting-guidelines/consort/) and ensure that all components of CONSORT are present in the manuscript, including how randomization was performed, allocation concealment, blinding of intervention, definition of lost to follow-up, power statement. When completing the checklist, please use section and paragraph numbers, rather than page numbers.

* Please report your abstract according to CONSORT for abstracts, following the PLOS Medicine abstract structure (Background, Methods and Findings, Conclusions) https://www.equator-network.org/reporting-guidelines/consort-abstracts/

* If your trial had to undergo important modifications in response to extenuating circumstances, please complete the CONSERVE-CONSORT checklist and provide in your Supporting Information; (https://www.equator-network.org/reporting-guidelines/guidelines-for-reporting-trial-protocols-and-completed-trials-modified-due-to-the-covid-19-pandemic-and-other-extenuating-circumstances-the-conserve-2021-statement/). When completing the checklist, please use section and paragraph numbers, rather than page numbers.

* In keeping with our commitment to Open Science, please include the study protocol document and analysis plan (including any amendments) as Supporting Information to be published with the manuscript if accepted.

* Please note that PLOS Medicine requires prospective, public registration of a data sharing plan (as part of mandatory clinical trials registration) for all clinical trials that began enrollment on or after January 1, 2019, in accordance with ICMJE requirements.

OBSERVATIONAL STUDIES

* Abstract: Please include the study design, population and setting, number of participants, years during which the study took place (enrollment and follow up), length of follow up, and main outcome measures.

* Please ensure that the study is reported according to the STROBE (or appropriate STOBE extension) guideline (available from: https://www.equator-network.org/reporting-guidelines/strobe) and include the completed STROBE (or STROBE extension) checklist as Supporting Information. Please add the following statement, or similar, to the Methods: "This study is reported as per the Strengthening the Reporting of Observational Studies in Epidemiology (STROBE) guideline (S1 Checklist)." When completing the checklist, please use section and paragraph numbers, rather than page numbers.

* [FOR POPULATION HEALTH/REGISTRY STUDIES] Please ensure that the study is reported according to the RECORD guideline (available from https://www.record-statement.org) and include the completed checklist as Supporting Information. Please add the following statement, or similar, to the Methods: "This study is reported as per the Reporting of Studies Conducted using Observational Routinely-Collected Data (RECORD) guideline (S1 Checklist)." When completing the checklist, please use section and paragraph numbers, rather than page numbers.

* [FOR POPULATION HEALTH ESTIMATES] Please ensure that the study is reported according to the GATHER statement (available from https://www.equator-network.org/reporting-guidelines/gather-statement) and include the completed checklist as Supporting Information. Please add the following statement, or similar, to the Methods: "This study is reported as per the Guidelines for Accurate and Transparent Health Estimates Reporting (GATHER) statement (S1 Checklist)." When completing the checklist, please use section and paragraph numbers, rather than page numbers.

* [FOR MEDIATION ANALYSES] We recommend that the study is reported according to the AGReMA statement (https://agrema-statement.org/#:~:text=AGReMA%20is%20an%20evidence%2D%20and,randomised%20trials%20and%20observational%20studies) and include the completed checklist as Supporting Information. Please add the following statement, or similar, to the Methods: "This study is reported as per the Guideline for Reporting Mediation Analyses (AGReMA) statement (S1 Checklist)." When completing the checklist, please use section and paragraph numbers, rather than page numbers.

* For all observational studies, in the manuscript text, please indicate: (1) the specific hypotheses you intended to test, (2) the analytical methods by which you planned to test them, (3) the analyses you actually performed, and (4) when reported analyses differ from those that were planned, transparent explanations for differences that affect the reliability of the study's results. If a reported analysis was performed based on an interesting but unanticipated pattern in the data, please be clear that the analysis was data driven.

* Please state in the Methods section whether the study had a prospective protocol or analysis plan. If a prospective analysis plan (from your funding proposal, IRB or other ethics committee submission, study protocol, or other planning document written before analyzing the data) was used in designing the study, please include the relevant document(s) with your revised manuscript as a Supporting Information file to be published alongside your study and cite it in the Methods section. A legend for this file should be included at the end of your manuscript. If no such document exists, please make sure that the Methods section transparently describes when analyses were planned, and when/why any data-driven changes to analyses took place. Changes in the analysis, including those made in response to peer review comments, should be identified as such in the Methods section of the paper, with rationale.

MODELLING STUDIES

The following list is derived from Geoffrey P Garnett, Simon Cousens, Timothy B Hallett, Richard Steketee, Neff Walker. Mathematical models in the evaluation of health programmes. (2011) Lancet DOI:10.1016/S0140-6736(10)61505-X:

* If pertinent, please provide a diagram that shows the model structure, including how the natural history of the disease is represented, the process and determinants of disease acquisition, and how the putative intervention could affect the system.

* Please provide a complete list of model parameters, including clear and precise descriptions of the meaning of each parameter, together with the values or ranges for each, with justification or the primary source cited and important caveats about the use of these values noted.

* Please provide a clear statement about how the model was fitted to the data, including goodness-of-fit measure, the numerical algorithm used, which parameter varied, constraints imposed on parameter values, and starting conditions.

* For uncertainty analyses, please state the sources of uncertainties quantified and not quantified [can include parameter, data, and model structure].

* Please provide sensitivity analyses to identify which parameter values are most important in the model. Uncertainty estimates seek to derive a range of credible results on the basis of an exploration of the range of reasonable parameter values. The choice of method should be presented and justified.

* Please discuss the scientific rationale for the choice of model structure and identify points where this choice could influence conclusions drawn. Please also describe the strength of the scientific basis underlying the key model assumptions.

* For studies that develop a prediction model or evaluate its performance, please ensure that the study is reported according to the TRIPOD statement (https://www.equator-network.org/reporting-guidelines/tripod-statement) and include the completed checklist as Supporting Information. Please add the following statement, or similar, to the Methods: "This study is reported as per the Transparent Reporting of a Multivariable Prediction Model for Individual Prognosis Or Diagnosis (TRIPOD) statement (S1 Checklist)." For studies using machine learning, please use the TRIPOD-AI checklist. When completing the checklist, please use section and paragraph numbers, rather than page numbers.

DIAGNOSTIC STUDIES

* Please ensure that the study is reported according to the STARD guideline (https://www.equator-network.org/reporting-guidelines/stard/) and include the completed STARD checklist as Supporting Information. Please add the following statement, or similar, to the Methods: "This study is reported as per the Standards for Reporting of Diagnostic Accuracy (STARD) guideline (S1 Checklist)." When completing the checklist, please use section and paragraph numbers, rather than page numbers.

* Please structure your Abstract according to STARD for Abstracts (https://www.equator-network.org/reporting-guidelines/stard-abstracts/).

* Please structure the Methods section using the following sub-headings: Study design, Participants, Test methods, Analysis.

* Please include a diagram to describe the flow of participants through the study (typically figure 1).

MENDELIAN RANDOMIZATION STUDIES

* Please ensure that the study is reported according to the STROBE-MR guideline (https://www.equator-network.org/reporting-guidelines/strobe/) and include the completed STROBE-MR checklist as Supporting Information. Please add the following statement, or similar, to the Methods: "This study is reported as per the Strengthening the Reporting of Observational Studies in Epidemiology (STROBE) guideline, specific for mendelian randomization (S1 Checklist)." When completing the checklist, please use section and paragraph numbers, rather than page numbers.

* In the Introduction, please describe the exposure and the evidence for a potential causal relationship between exposure and outcome.

* In the Methods, please explicitly state the 3 core instrumental variable assumptions for the main analysis (relevance, independence, and exclusion restriction), as well assumptions for any additional or sensitivity analysis.

* In the Methods, please describe the MR estimator (e.g., 2-stage least squares, Wald ratio) and related statistics. Detail the included covariates and, in case of 2-sample MR, whether the same covariate set was used for adjustment in the 2 samples.

* If you are presenting an instrumental variable estimate, please compare this to the conventional observational estimate.

* Report the associations between genetic variant and exposure and between genetic variant and outcome, preferably on an interpretable scale.

* Report MR estimates of the relationship between exposure and outcome and the measures of uncertainty from the MR analysis, on an interpretable scale, such as odds ratio or relative risk per SD difference.

* If relevant, please consider translating estimates of relative risk into absolute risk for a meaningful time period.

* Please consider including plots to visualize results (e.g., forest plot, scatterplot of associations between genetic variants and outcome vs between genetic variants and exposure).

SURVEY-BASED STUDIES

* Please ensure that the study is reported according to the CROSS guideline (https://www.equator-network.org/reporting-guidelines/a-consensus-based-checklist-for-reporting-of-survey-studies-cross/) and include the completed CROSS checklist as Supporting Information. Please add the following statement, or similar, to the Methods: "This study is reported as per A Consensus-Based Checklist for Reporting of Survey Studies (CROSS) guideline (S1 Checklist)." When completing the checklist, please use section and paragraph numbers, rather than page numbers.

* Please report your survey response rates according to AAPOR recommendations (https://aapor.org/standards-and-ethics/best-practices/)

* Please define how the population surveyed was sampled.

* Please compare characteristics of respondents and nonrespondents if possible.

* If sequential waves of the survey were sent, please specify whether the characteristics of respondents changed over time or remained constant.

* Please include the survey response rate in the Abstract.

* Please include a copy of the survey in the supplementary files.

SYSTEMATIC REVIEWS & META-ANALYSES

* Please report your SR/MA according to the PRISMA guidelines provided at the EQUATOR site. http://www.equator-network.org/reporting-guidelines/prisma/. Please provide the completed PRISMA checklist as Supporting Information. When completing the checklist, please use section and paragraph numbers, rather than page numbers. Please add the following statement, or similar, to the Methods: "This study is reported as per the Preferred Reporting Items for Systematic Reviews and Meta-Analyses (PRISMA) guideline (S1 Checklist)."

* Abstract: Please report your abstract according to PRISMA for abstracts (https://doi.org/10.1371/journal.pmed.1001419) following the PLOS Medicine abstract structure (Background, Methods and Findings, Conclusions). Please ensure you provide dates of search, data sources, number of studies included, types of study designs included, eligibility criteria, and synthesis/appraisal methods.

* Please note that we expect searches to be updated to within 6 months of the time of submission.

QUALITATIVE STUDIES

* Please report your qualitative study according to the appropriate study design provided at (http://www.equator-network.org/?post_type=eq_guidelines&eq_guidelines_study_design=qualitative-research&eq_guidelines_clinical_specialty=0&eq_guidelines_report_section=0&s=) and provide the relevant completed checklist as a supplemental file. In the checklist, please include sufficient text excerpted from the manuscript to explain how you accomplished all applicable items. When completing checklists, please use section and paragraph numbers, rather than page numbers.

* We recommend that authors use the COREQ checklist, or other relevant checklists listed by the Equator Network, such as the SRQR, to ensure complete reporting (see: http://www.equator-network.org/?post_type=eq_guidelines&eq_guidelines_study_design=qualitative-research&eq_guidelines_clinical_specialty=0&eq_guidelines_report_section=0&s=). Please add the following statement, or similar, to the Methods: "This study is reported as per the Consolidated criteria for reporting qualitative research (COREQ): a 32-item checklist for interviews and focus groups (S1 Checklist)."

* In general, we expect qualitative studies to include the following: 1) defined objectives or research questions; 2) description of the sampling strategy, including rationale for the recruitment method, participant inclusion/exclusion criteria and the number of participants recruited; 3) detailed reporting of the data collection procedures; 4) data analysis procedures described in sufficient detail to enable replication; 5) a discussion of potential sources of bias; and 6) a discussion of limitations.

HEALTH ECONOMICS / COST-EFFECTIVENESS STUDIES

* Please ensure that the study is reported according to the CHEERS guideline (available from: https://www.equator-network.org/reporting-guidelines/cheers) and include the completed checklist as Supporting Information. Please add the following statement, or similar, to the Methods: "This study is reported as per the Strengthening the Consolidated Health Economic Evaluation Reporting Standards 2022 (CHEERS 2022) Statement (S1 Checklist)." When completing the checklist, please use section and paragraph numbers, rather than page numbers.

---

## [Decision Letter · Decision Letter 2]

25 Apr 2025

Dear Dr. Molaro,

Thank you very much for re-submitting your manuscript "The potential impact of declining development assistance for healthcare on population health: projections for Malawi" (PMEDICINE-D-24-03289R2) for review by PLOS Medicine.

I have discussed the paper with my colleagues and the academic editor and it was also seen again by 3 reviewers. I am pleased to say that provided the remaining editorial and production issues are dealt with we are planning to accept the paper for publication in the journal.

As you can see from the reviews, reviewer #1 has some remaining concerns. From his concerns, we would like you to address his comment 'e' and 'f' in the manuscript. We do appreciate you may only be able to do so textually.

Furthermore, we have some editorial issues that needs addressing. These issues are listed at the end of this email. Any accompanying reviewer attachments can be seen via the link below. Please take these into account before resubmitting your manuscript:

[LINK]

We look forward to receiving the revised manuscript by May 02 2025 11:59PM.   

Sincerely,

Suzanne De Bruijn, PhD

Associate Editor 

PLOS Medicine

plosmedicine.org

Requests from Editors:

*Please format your abstract to PLOS requirements.

*Please modify the structure of the discussion, to remove the bullet points and subheadings.

*There are several instances in the manuscript where there are claims of novelty, please remove these (for more guidelines, see below)

Further editorial requests:

* At this stage, we ask that you include a short, non-technical Author Summary of your research to make findings accessible to a wide audience that includes both scientists and non-scientists. The Author Summary should immediately follow the Abstract in your revised manuscript. This text is subject to editorial change and should be distinct from the scientific abstract. Ideally each sub-heading should contain 2-3 single sentence, concise bullet points containing the most salient points from your study. In the final bullet point of ‘What Do These Findings Mean?’ Please include the main limitations of the study in non-technical language.

Please see our author guidelines for more information: https://journals.plos.org/plosmedicine/s/revising-your-manuscript#loc-author-summary.

* Please confirm that your title complies with to PLOS Medicine's style. Your title must be nondeclarative and not a question. It should begin with main concept if possible. "Effect of" should be used only if causality can be inferred, i.e., for an RCT. Please place the study design ("A randomized controlled trial," "A retrospective study," "A modelling study," etc.) in the subtitle (ie, after a colon).

* Please confirm that your abstract complies with our requirements, including providing all the information relevant to this study type https://journals.plos.org/plosmedicine/s/submission-guidelines#loc-abstract

* Please ensure that the Introduction ends with a clear description of the study question or hypothesis.

* Please ensure that all abbreviations are defined at first use throughout the text.

* Please confirm that all numbers presented in the abstract are present and identical to numbers presented in the main manuscript text.

* Please review your text for claims of novelty or primacy (e.g. 'for the first time') and remove this language. In addition, please check that any use of statistical terms (such as trend or significant) are supported by the data, and if not please remove them.

* Please remove the 'conclusions' subheading.

* Please define all elements of box plots in the figure caption - center line, box limits and whiskers.

Comments from Reviewers:

Reviewer #1:

a) The manuscript mentions confidence intervals but does not explicitly state whether parametric or non-parametric methods were used to generate them.

b) The study conducted sensitivity analyses on consumable availability, but incorporating additional scenario testing (e.g., simulating dynamic GDP growth rates over time) would enhance the study's robustness.

c) Assessing extreme-case scenarios—such as economic downturns or donor funding reductions beyond IHME projections—could provide deeper insights into the resilience of the model's findings.

d) While the model is calibrated to historical data, it does not explicitly assess predictive validity. The authors could strengthen their analysis by employing a back-testing approach, comparing past projections with actual outcomes (if available). Additionally, external validation using alternative datasets (e.g., WHO health expenditure and outcome trends in comparable countries) could improve the credibility of their projections.

e) The model does not appear to account for the broader economic impacts of health funding changes, such as effects on labour productivity, poverty reduction, or private-sector investment in healthcare.

f) The authors assume constant health expenditure growth rates and fixed intervention effectiveness over time. However, real-world factors—such as medical inflation, technology adoption, and policy shifts—could significantly influence outcomes.

g) While the study estimates the health burden under different expenditure scenarios, incorporating an explicit cost-effectiveness analysis (e.g., cost per DALY averted) would enhance the policy relevance of the findings.

Reviewer #3:

The updated manuscript satisfactorily addresses all comments and questions

[LINK]

---

## [Editor Report · Decision Letter 3]

14 May 2025

Dear Dr. Molaro,

Thank you very much for re-submitting your manuscript "The potential impact of declining development assistance for healthcare on population health in Malawi: a modelling study" (PMEDICINE-D-24-03289R3) for review by PLOS Medicine.

I have discussed the paper with my colleagues and the academic editor. I am pleased to say that we are happy that all the scientific concerns are addressed. However, we have a few remaining editorial requests, which you can find at the end of this email. Provided the remaining editorial and production issues are dealt with we are planning to accept the paper for publication in the journal.

[LINK]

We look forward to receiving the revised manuscript by May 21 2025 11:59PM.   

Sincerely,

Suzanne De Bruijn, PhD

Associate Editor 

PLOS Medicine

plosmedicine.org

Requests from Editors:

*Regarding your query how to deal with the reviewer query asking for a comparative analysis: I would suggest to leave the statement, but modify it to: To our knowledge, no other study has directly linked future healthcare spending implications to population health outcomes in Malawi or other countries.” Ie, replace ‘At present’ with ‘To our knowledge’.

*Abstract, 1st sentence of Methods and Findings section, please indicate the meaning of ‘this’; eg, “…to estimate the impact *that declining DHA* could have on health system capacities…”

*Abstract, Methods and Findings, I think the word ‘if’ needs to be removed from the following sentence, “The burden due to non-communicable diseases, on the other hand, is found to increase irrespective of yearly growth in health expenditure, if assuming current reach and scope of intervention.”

*Abstract, please add a sentence to the end of the Methods and Findings section to indicate the major limitation(s) of the study/methodology.

* In the author summary, in the final bullet point of 'What Do These Findings Mean?', please include the main limitations of the study in non-technical language.

*Please remove the bullet points at the end of the introduction. Instead, incorporate these sentences into a final paragraph in the introduction.

*Please delete all numbering for the sections, as well as references to these (e.g. in the discussion there is a statement 'as discussed in section 3.2'). Subheadings should be text only.

*Please define all abbreviations at first use. There are some that are only defined at a later point in the manuscript (e.g. HIV/TB are used in the introduction, but defined in the methods). In addition, if an abbreviation is used in the abstract, author summary and main text, please ensure that it is defined in the main text (and also in author summary/abstract, IF it is used more than once in these sections; if only used once, please spell it out).

*Thank you for removing the claims of novelty. However, we found one in the discussion (page 14); Could you please adapt this sentence to “In conclusion, this analysis is the first, to our knowledge, to quantify the potential overall consequences…”

*The first sentence of the funding statement is slightly unclear. Please modify to: “This project is funded by The Wellcome Trust (223120/Z/21/Z), which also contributed to the salaries…” (or similar)

*Thank you for stating all the data is accessible, and providing a link to the model on Github. However, because Github depositions can be readily changed or deleted, we encourage you to make a permanent DOI'd copy (e.g. in Zenodo) and provide the URL. Please review our guidelines at https://journals.plos.org/plosmedicine/s/materials-software-and-code-sharing and ensure that your code is shared in a way that follows best practice and facilitates reproducibility and reuse.

* Statistical reporting: Please revise throughout the manuscript, including tables and figures.

- Please report statistical information as follows to improve clarity for the reader ""22% (95% CI [13,28]; p</=)"".

- Please separate upper and lower bounds with commas instead of hyphens as the latter can be confused with reporting of negative values.

- Please repeat statistical definitions (HR, CI etc.) for each set of parentheses.

---

## [Editor Report · Decision Letter 4]

30 May 2025

Dear Dr Molaro, 

On behalf of my colleagues and the Academic Editor, Peter MacPherson, I am pleased to inform you that we have agreed to publish your manuscript "The potential impact of declining development assistance for healthcare on population health in Malawi: a modelling study" (PMEDICINE-D-24-03289R4) in PLOS Medicine.

Please also note that the study limitations should be communicated explicitly, using the language "The main limitations of the study include..." both in the Abstract and the Author summary. This can be done after acceptance. We of course appreciate that you have already outlined the limitations, but they are not signposted as such, so we ask that this change is made for maximal clarity.

PRESS

Sincerely, 

Suzanne De Bruijn, PhD 

Senior Editor 

PLOS Medicine